# A Mesozoic clown beetle myrmecophile (Coleoptera: Histeridae)

Yu-Lingzi Zhou[1,2], Adam Ślipiński[2], Dong Ren[3], Joseph Parker[4]*

[1]Key Laboratory of Zoological Systematics and Evolution, Institute of Zoology, Chinese Academy of Sciences, Beijing, China; [2]Australian National Insect Collection, CSIRO, Canberra, Australia; [3]College of Life Sciences, Capital Normal University, Beijing, China; [4]Division of Biology and Biological Engineering, California Institute of Technology, Pasadena, United States

**Abstract** Complex interspecies relationships are widespread among metazoans, but the evolutionary history of these lifestyles is poorly understood. We describe a fossil beetle in 99-million-year-old Burmese amber that we infer to have been a social impostor of the earliest-known ant colonies. *Promyrmister kistneri* gen. et sp. nov. belongs to the haeteriine clown beetles (Coleoptera: Histeridae), a major clade of 'myrmecophiles'—specialized nest intruders with dramatic anatomical, chemical and behavioral adaptations for colony infiltration. *Promyrmister* reveals that myrmecophiles evolved close to the emergence of ant eusociality, in colonies of stem-group ants that predominate Burmese amber, or with cryptic crown-group ants that remain largely unknown at this time. The clown beetle-ant relationship has been maintained ever since by the beetles host-switching to numerous modern ant genera, ultimately diversifying into one of the largest radiations of symbiotic animals. We infer that obligate behavioral symbioses can evolve relatively rapidly, and be sustained over deep time.
DOI: https://doi.org/10.7554/eLife.44985.001

## Introduction

A pervasive feature of colony-forming insect societies is the profusion of intruder arthropods that have evolved to exploit their rich resources (*Kistner, 1979*; *Kistner, 1982*; *Hölldobler and Wilson, 1990*; *Parker, 2016*). The diversity of such organisms is impressive, with ~10,000 species hypothesized to target or profit from ant nests alone (*Elmes, 1996*). Hostility of ant workers to virtually all non-nestmate organisms has selected for defensive or host-deceptive adaptations in myrmecophiles which are often phenotypically remarkable, involving changes in anatomy, chemical ecology and behavior (*Kistner, 1979*; *Kistner, 1982*; *Hölldobler and Wilson, 1990*; *Parker, 2016*). In a number of cases, traits have arisen that enable the myrmecophile to manipulate worker behavior, circumventing aggression and enabling social interactions to evolve that assimilate the symbiont into colony life. Such relationships rank among the most behaviorally intimate interactions known between animal species (*Kistner, 1979*; *Hölldobler and Wilson, 1990*; *Parker, 2016*), and are typically achieved by the myrmecophile's capacity to mimic the chemical and/or tactile cues involved in nestmate recognition (*Kistner, 1979*; *Hölldobler and Wilson, 1990*; *Parker, 2016*). The clown beetle family Histeridae includes multiple lineages that have independently evolved myrmecophily (*Parker, 2016*; *Kovarik and Caterino, 2005*), including Haeteriinae, a subfamily of ~335 described species comprising possibly the single largest radiation of myrmecophiles known within the Coleoptera (*Parker, 2016*; *Kovarik and Caterino, 2005*; *Helava et al., 1985*). We report the discovery of a crown-group haeteriine in Upper Cretaceous Burmese amber, revealing that the clown beetle-ant interaction has an exceptionally deep evolutionary history. To our knowledge, the relationship constitutes the most ancient behavioral symbiosis known in the Metazoa.

*For correspondence: joep@caltech.edu

Competing interests: The authors declare that no competing interests exist.

**eLife digest** Many animals live lives that are closely intertwined with those of other species. While a clown fish sheltering within the tentacles of a sea anemone may be a textbook example, 'symbiotic' interactions that occur inside ant nests are among some of the most dramatic.

Known as myrmecophiles – after the Greek for 'ant lovers', many insects, spiders and mites have evolved to live alongside ants in one way or another. Some of these animals display elaborate behaviors – like mouth-to-mouth feeding or grooming of worker ants – which assimilates them into the nest society; some even release chemicals that mimic the ants' own scents to avoid being detected as an intruder.

The earliest examples of ancestral ants are found encapsulated in 99-million-year-old amber from a mine in northern Myanmar (Burma). Zhou et al. have now discovered an ancient beetle, perfectly preserved in the same amber deposits, that may have also lived within the colonies of those earliest-known ants. Based on its appearance, the beetle – named *Promyrmister kistneri* – belongs within a subfamily of clown beetles (called the Haeteriinae) that are all specialized nest intruders with dramatic behavioral and chemical adaptations that help them to infiltrate ant colonies.

The ancient clown beetle shares several of features with its modern relatives – including thick, spiked legs and well-protected head and antennae – which are believed to help the beetles withstand handling by the ants' jaws. The specimen also has glands near the base of its legs, implying that it also released chemical signals that may have helped it to deceive or pacify the ancient ants.

The fact that this extinct clown beetle is as old as the earliest-known ants implies that the close relationship between these insects has been sustained for an exceptionally long time. It is potentially the oldest known example of a symbiotic interaction in the animal kingdom that depends on social interactions between the two organisms. However, the host ants of *Promyrmister* are believed to be long-extinct, suggesting that symbiotic clown beetles had to switch to living inside colonies of modern ants to circumvent their own extinction. This flexibility to adapt to new partner species may be a critical feature that allows some symbiotic organisms to persist throughout evolution.
DOI: https://doi.org/10.7554/eLife.44985.002

## Results and discussion

### Systematic palaeontology

Order Coleoptera Linnaeus, 1758
Superfamily Histeroidea Gyllenhal, 1808
Family Histeridae Gyllenhal, 1808
Subfamily Haeteriinae Marseul, 1857
***Promyrmister kistneri* Zhou, Ślipiński and Parker gen. et sp. nov.**

### Holotype

Sex unknown. CNU-008021, deposited in Key Laboratory of Insect Evolution and Environmental Changes, Capital Normal University, Beijing. The holotype is well preserved in a small, transparent amber piece, 5.5 mm length ×3.5 mm width (*Figure 1—figure supplement 1A*). The entire external anatomy is observable (*Figure 1A–C*), but the left region of the dorsal side is partially covered by white exudate (*Figure 1A,G*) emanating from the ventral side of the pronotal margin (arrow in *Figure 1A,G*).

### Diagnosis of new genus and species

Haeteriine histerid that is distinguished from all other genera and species of Haeteriinae by possession of the following combination of characters: (1) deep depression behind meso- and metacoxae (*Figure 1C,H*); (2) metaventral postcoxal line recurved and extending laterally to metanepisternum (*Figure 1B,C*; (3) three complete striae on each elytron (*Figure 1A*); (4) lack of dorsal furrows on pronotum demarcating glandular lobe (*Figure 1A,B*); (5) strongly developed apical spur on protibia

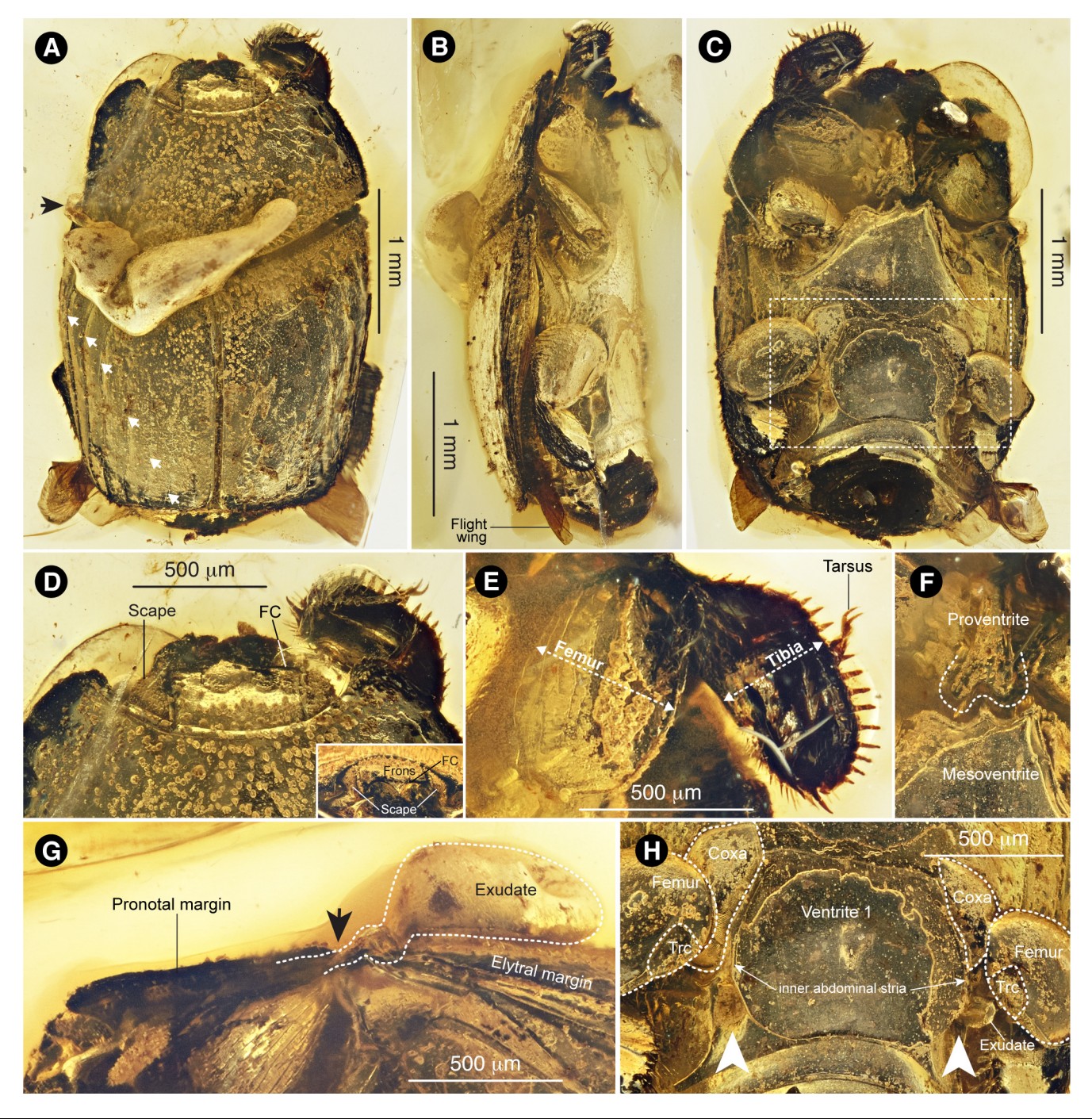

**Figure 1.** *Promyrmister kistneri* gen.et sp. nov. (**A**) Dorsal habitus of holotype CNU-008021 with origin of exudate globule (black arrow) and elytral striae (white arrows) indicated; the three lateral striae are complete (top three arrows), the three medial striae appear incomplete. (**B**) Right lateral habitus with flight wing indicated. (**C**) Ventral habitus, boxed region expanded in panel H. (**D**) Head, dorsal and (inset) frontal views, with antennal scapes and frontoclypeal carina (FC) indicated. (**E**) Right foreleg, laterally expanded femur and tibia indicated. (**F**) Proventral-mesoventral boundary showing proventral keel with posterior incision. (**G**) Left pronotal margin, lateral view, showing possible origin of putative glandular exudate; arrow in G corresponds to that in panel A. (**H**) Ventrite one showing proximal leg segments (Trc: trochanter) and postcoxal gland openings (white arrows), with globule of putative exudate emanating from left postcoxal gland opening.

DOI: https://doi.org/10.7554/eLife.44985.003

The following figure supplements are available for figure 1:

**Figure supplement 1.** Amber piece and frontal view of holotype of Promyrmister kistneri sp.nov.

*Figure 1 continued on next page*

*Figure 1 continued*

DOI: https://doi.org/10.7554/eLife.44985.004

**Figure supplement 2.** Further morphology of *Promyrmister*.

DOI: https://doi.org/10.7554/eLife.44985.005

(*Figure 1D,E*); (6) frontoclypeus carinate medially (*Figure 1D*, *Figure 1—figure supplement 1B,C*); (7) triangular-shaped cavities to receive scapes of antennae (*Figure 1—figure supplement 1B,C*); (8) glandular opening in postcoxal cavity behind metacoxae (*Figure 1H*; *Figure 1—figure supplement 2A,B,D*). *Promyrmister* specifically differs from the closely related *Haeterius* in having deep epistomal depressions (*Figure 1—figure supplement 1B,C*; compare to *Figure 2—figure supplement 1D*), carinated epistomal striae convergent in the middle (*Figure 1—figure supplement 1B,C*) and paddle-shaped protibia with large apical spur (*Figure 1D,E*).

## Locality and age

The holotype inclusion is derived from an amber mine located near Noije Bum, Tanaing, Kachin, Myanmar. The U-Pb dating of zircons from the volcanoclastic matrix yielded an age of $98.79 \pm 0.62$ million years (*Shi et al., 2012*).

## Etymology

The generic name *Promyrmister* is a combination of the Greek πϱοˊ (pro) meaning 'before' or 'early', μῦϱμηξ (myrmex) meaning 'ant', and *Hister* Linnaeus, type genus of Histeridae. The name refers to the likely symbiotic habits of the fossil taxon inside early ant colonies. The gender is masculine. The specific epithet recognizes the lifetime contribution of Dr. David H. Kistner, a global authority on social insect symbionts.

## Description

Length 3.2 mm, width 2.3 mm. Body elongate oval (*Figure 1A,C*), moderately convex (*Figure 1B*); black or dark brown with dorsal surfaces bearing short and somewhat squamiform setae, visible along pronotal and elytral edges but on dorsal side often obscured by accumulation of water/dirt and appearing as tiny granules; interstices between setae moderately to distinctly shiny.

Head only partially visible, deeply inserted into prothorax (*Figure 1D*; *Figure 1—figure supplement 1B*). Frons with distinct frontoclypeal carina (FC in *Figure 1D*), and widely interrupted frontal stria, the lateral parts of which extend to the frontoclypeal carina, connecting to inwardly-arching epistomal striae (*Figure 1—figure supplement 1C*). Clypeus bordered by deep epistomal depressions to receive antennal scapes in repose (*Figure 1—figure supplement 1B*). Clypeus and labrum apparently fused but with distinct transverse ridge above the base of labrum. Mandibles strongly arcuate apically. Antennal scape large and triangular (*Figure 1—figure supplement 1B*), covering eye, and densely rugose dorsally.

Prothorax (length 1.0 mm and width 1.8 mm) widest at base, sides weakly rounded, converging anteriorly, anterior angles distinctly projecting and rounded; posterior angles weakly obtuse (*Figure 1A*). Lateral margins crenulate, each projection bearing weakly squamiform setae. Pronotum with marginal stria complete anteriorly and along lateral margins; sides without impressions or obvious gland openings along lateral carina; disc weakly convex, setose. Proventral lobe strongly prominent medially, covering most of ventral head surface and extending laterally to antennal cavities without visible marginal stria. Prosternal process (proventrite) narrowly elevated with apex about 0.1 times as broad as prothorax, expanding apically and deeply emarginate at apex (*Figure 1F*); prosternal carinae converging anteriorly but not apparently joined; junction between proventrite and prosternal lobe deeply depressed. Antennal cavity present on anterior angles of hypomeron, deep and completely closed from below via proventral alae (*Figure 1—figure supplement 1B*). Procoxae not clearly visible. Trochanter large, triangular and bearing several long setae; profemur very broad, width nearly 0.45 mm, and flat, covering most of the ventral side of prothorax (*Figure 1E*); protibia flat and expanded, width about 0.38 mm, bearing row of strong spines along external edge and an apical spur (*Figure 1E*); protarsi short and thin, sitting in straight groove on dorsal side of protibia (*Figure 1D*). Scutellum obscured dorsally by secretion (*Figure 1A*). Elytra (1.8 mm length ×2.3 mm

width), with relatively complete dorsal striae 1–three and reduced striae 4–6 (white arrows in *Figure 1A*); outer subhumeral stria complete, sutural stria very fine and visible apically.

Mesoventrite between mesocoxae very broad, about 1/3 of body width at the same position (*Figure 1C*); anterior margin projecting medially fitting into prosternal process (*Figure 1F*); Margin between metaventrite and visible abdominal ventrite one shallowly grooved. Ventrite one weakly convex medially, deeply concave laterally (*Figure 1C*) to accommodate strongly flattened legs. Post-coxal lines behind meso- and metacoxae completely recurved dorsally; discrimen complete. Hind coxae triangular (*Figure 1C,H*; *Figure 1—figure supplement 2A–C*), large and broadly separated from each other, with numerous regular rows of oblique striae on inner surface (*Figure 1—figure supplement 2A–C*); hind femur oval-shaped, distinctly large and flat, enveloping small and setose trochanter (*Figure 1C,H*). Hind tibia flat, paddle-like, about as broad as femur (*Figure 1C*), with a row of strong spines along external edge, double rows of stiff and apically modified setae along inner edge, inner surface with transverse ridges and row of pointed spines before; apical spur small. Tarsal formula 5-5-5. Hind tarsi slim, locating on inner side of tibia, tarsomere one longest, length subequal to tarsomeres 2–four combined. Wings visible apically (*Figure 1B*), presumed functional.

Abdomen with median part of ventrite one delimited to a flat and polished central plate by inner abdominal stria, much longer than the remaining ventrites combined (*Figure 1C*); postcoxal line (outer abdominal stria) recurved and strongly diverging laterally (*Figure 1B,C*); also with deeply depressed rest for hind tibia outside the postcoxal line. Large abdominal gland opening behind hind coxa located between inner and outer abdominal striae (*Figure 1H*), on right side with exudate flowing out (paired arrows in *Figure 1H*; *Figure 1—figure supplement 2B,D*, note that laser reflectance indicates this is solidified material and not a gas/air bubble, which would appear dark). Ventrites 2–four equal in length, without posterior marginal striae.

## Systematic position

The fossil beetle is placed in Histeridae based on its possession of the following characters (*Kovarik and Caterino, 2000*): (i) a broad and compact body shape (*Figure 1A,C*); (ii) striate elytra that expose the two posterior abdominal tergites (*Figure 1A*); ii) five visible abdominal sternites (*Figure 1C*); (iv) short legs with broad, flattened tibiae (*Figure 1C,E*); (v) antenna short and geniculate with compact, 3-segmented club (*Figure 1—figure supplement 1B*); (vi) antenna retracting into cavity underneath the pronotum (*Figure 1—figure supplement 1B*); (vii) tarsal formula 5-5-5. Of the 11 histerid subfamilies (*Bouchard et al., 2011*), *Promyrmister* can be placed unequivocally in the subfamily Haeteriinae based on the following characters: head deflexed, with clypeus arched downwards in a different plane to the vertex (*Figure 1D*); antenna with enlarged, triangular scape received in repose in frontal groove and hiding the eye (*Figure 1D*; *Figure 1—figure supplement 1B,C*); antennal cavities located on hypomeron (*Figure 1—figure supplement 1B*), covered from below by proventral alae; proventral lobe strongly developed anteriorly, covering head from below, and extending laterally to form proventral alae (without lateral notch); proventral keel narrowly elevated between coxae and distinctly emarginate posteriorly to receive the projecting mesoventral process (*Figure 1F*). Additionally, the front, mid, and hind legs are extremely broad (*Figure 1C,E*), which is a feature of the clade *Yarmister* + Haeteriinae (*Caterino and Tishechkin, 2015*), in which *Yarmister* lacks the distinctly elevated proventral keel with posterior incision, which is present in *Promyrmister* and is an autapomorphy of the Haetaeriinae (*Helava et al., 1985*). The labrum of the fossil specimen may also be fused, but fossilization position precludes definitive assessment.

Within Haeteriinae, the new taxon appears to bear a close relationship to the genus *Haeterius* Erichson and some closely allied genera that share several morphological characters supporting their monophyly (*Yélamos, 1997*), principally the broad and externally rounded tibiae, the deep depressions behind meso- and metacoxae to accommodate retracted legs (*Figure 1C,H*) (*Caterino and Tishechkin, 2015*), the metaventral postcoxal line being recurved and extending laterally to the metanepisternum (*Figure 1B,C*), and the presence of three complete striae on each elytron (*Figure 1A*). Morphological features of the extant *Haeterius*, with key character states shared with *Promyrmister*, are shown in *Figure 2—figure supplement 1* (see Diagnosis for separation of *Promyrmister* and *Haeterius*). Consistent with our evaluation of *Promyrmister*'s likely phylogenetic placement, both cladistic and Bayesian analysis of a set of morphological characters (*Caterino and Tishechkin, 2015*) from the fossil specimen and a selection of Recent histerid taxa places the new

taxon within Haeteriinae as sister to *Haeterius* (*Figure 2A,B*; *Figure 2—figure supplements 2* and *3*). We infer that *Promyrmister* represents an extinct Cretaceous lineage that belongs within the crown-group of Haeteriinae.

## *Promyrmister* and deep-time persistence of a social symbiosis

Symbiotic relationships in which different animal species interact socially with each other have arisen sporadically across the metazoan tree of life. Such relationships encompass a spectrum of dependency, from transient, facultative associations seen in mixed-species groups of insectivorous birds (*Sridhar et al., 2009*), cetaceans and ungulates (*Stensland et al., 2003*; *Goodale et al., 2017*), to obligate symbiotic lifestyles typified by brood parasitic cuckoos, cowbirds (*Johnsgard, 1997*), mutualistic cleaner fish (*Grutter, 1999*) and oxpeckers (*Nunn et al., 2011*). Of all animal groups, however, the complex societies of ants play host to the greatest diversity of behavioral symbionts. Several major radiations of myrmecophiles are known, each containing hundreds of symbiotic species, including the lycaenid butterflies (*Pierce et al., 2002*), eucharitid wasps (*Murray et al., 2013*), paussine ground beetles (*Moore and Robertson, 2014*) and multiple lineages of rove beetles (*Kistner, 1979*; *Kistner, 1982*; *Parker, 2016*; *Parker and Grimaldi, 2014*; *Seevers, 1965*; *Maruyama and Parker, 2017*). The diversity and often-broad geographic ranges of these clades imply that their relationships with ants are evolutionarily ancient (*Parker and Grimaldi, 2014*; *Yamamoto et al., 2016*). Although fossil myrmecophiles are known from as far back as the Eocene (*Parker and Grimaldi, 2014*; *Wasmann, 1929*), ant eusociality is known to be at least twice as old,

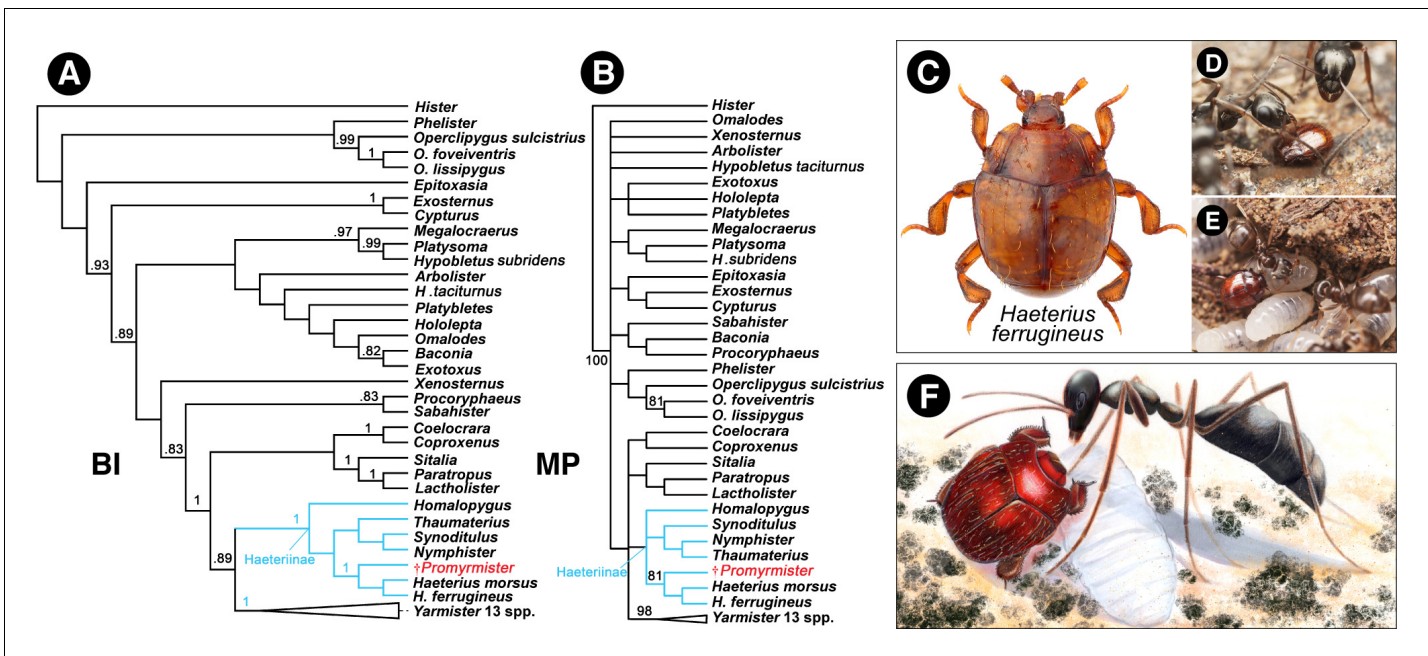

**Figure 2.** Phylogenetic relationships of *Promyrmister*. (A) Consensus Bayesian Inference (BI) tree of representative histerid taxa including Haeteriinae, and *Promyrmister*. Posterior probabilities above 0.8 are shown on branches. (B) The consensus parsimony tree (MP) using TNT under the Traditional Search; bootstrap percentages above 80 are shown on branches. (C) Habitus photograph of *Haeterius* (*H. ferrugineus*), an inferred extant close relative of *Promyrmister* (photo credit: C. Fägerström) (D, E) Living *Haeterius ferrugineus* beetles interacting with *Formica* (D) and *Lasius* (E) host ants (photo credit: P. Krásenský). (F) Reconstruction of *Promyrmister* with stem-group host ant and larva (ant based on *Gerontoformica*).
DOI: https://doi.org/10.7554/eLife.44985.006

The following figure supplements are available for figure 2:

**Figure supplement 1.** Scanning electron micrographs of extant *Haeterius ferrugineus* (Olivier, 1789).
DOI: https://doi.org/10.7554/eLife.44985.007

**Figure supplement 2.** Part A of single tree obtained by parsimony analysis of morphological matrix using implied weighting (K = 13) in TNT.
DOI: https://doi.org/10.7554/eLife.44985.008

**Figure supplement 3.** Part B of single tree obtained by parsimony analysis of morphological matrix using implied weighting (K = 13) in TNT.
DOI: https://doi.org/10.7554/eLife.44985.009

with the earliest definitively social ants occurring in Upper Cretaceous Burmese amber (*Barden and Grimaldi, 2016*). Whether their colonies were targeted by myrmecophiles has, however, been unclear: ants are comparatively scarce in Cretaceous ambers (*Grimaldi and Agosti, 2000*; *LaPolla et al., 2013*; *Barden, 2016*; *Barden, 2018*), and myrmecophilous invertebrates typically live at densities orders of magnitude lower than their hosts (*Kistner, 1979*). The unlikely discovery of a myrmecophile clown beetle in Burmese amber reveals that a major radiation of ant symbionts has its origins in Mesozoic ant societies.

Analysis of *Promyrmister*'s morphology and phylogenetic position indicates the new genus represents an extinct lineage within the crown-group of Haeteriinae, a clade of obligate myrmecophiles (*Figure 2A,B*; *Figure 2—figure supplements 2* and *3*; see Materials and methods). In haeteriine taxa for which detailed behavioral observations exist, the beetles have been shown to engage in intimate behaviors with ants, involving stomodeal trophallaxis (mouth-to-mouth feeding) (*Wheeler, 1908*; *Henderson and Jeanne, 1990*; *Akre, 1968*), grooming workers with their appendages (and being groomed or licked by hosts in return) (*Akre, 1968*), physically grasping onto ants (phoresis) (*Akre, 1968*; *von Beeren and Tishechkin, 2017*), or being carried around nests by workers (*Kistner, 1982*). Mimicry of colony cuticular hydrocarbons occurs (*Lenoir et al., 2012*), as well as chemical manipulation of host ants via 'appeasement' substances exuded from gland openings on the margins of the prothorax (*Kistner, 1982*; *Seyfried, 1928*) or in the postcoxal regions of the beetle's underside (*Figure 2—figure supplement 1C*). *Promyrmister* appears to be closely allied to the extant genus *Haeterius* (*Figure 2A–C*; *Figure 2—figure supplement 1*). This genus and a handful of closely related taxa including *Eretmotus*, *Sternocoelis* and *Satrapes* comprise the only group of Haeteriinae known to occur in the Palaearctic, consistent with the Eurasian palaeolocality of *Promyrmister* in Burmese amber. Like all of these genera, *Promyrmister* exhibits classical haeteriine attributes that are thought to be true adaptations for myrmecophily, including broad expansions of the tibiae with spines on the outer margin (*Figure 2C*; *Figure 2—figure supplement 1E*), short tarsi received on the outer face of each tibia (*Figure 2—figure supplement 1E*), a triangular antennal scape (*Figure 2—figure supplement 1D*), pronounced antennal cavities on the prothoracic hypomeron (*Figure 2—figure supplement 1D*) and a broad proventral lobe to fully embrace the retracted head (*Figure 2—figure supplement 1B–D*). Those features are thought to be protective modifications that enable myrmecophile beetles to withstand handling by ant mandibles (*Parker, 2016*).

We and others have previously described rove beetles (Staphylinidae) in the Burmese palaeofauna that were putative symbionts of termite colonies (*Yamamoto et al., 2016*; *Cai et al., 2017*). These specimens exhibit a defensive ecomorphology and are thought to have been persecuted intruders that were not behaviorally integrated into their host's societies (*Yamamoto et al., 2016*; *Cai et al., 2017*). In contrast, Haeteriinae embody a form of true behavioral symbiosis, where the relationship with host ants can involve social interactions (*Figure 2D,E*; Figure 4). The prothoracic glandular openings of Haeteriinae that secrete putative appeasement compounds are challenging to demonstrate even in extant taxa, but in the *Promyrmister* holotype, a large globule of possible exudate originates from the left margin of the prothorax, consistent with the position of such glands (*Figure 1A,G*). Additionally, *Promyrmister* possesses clear postcoxal secretory glands (*Figure 1H*; *Figure 1—figure supplement 2A,B,D*), with a globule of possible exudate emanating from the postcoxal gland opening on the right side of the body (*Figure 1H*; *Figure 1—figure supplement 2B,D*). Beyond *Promyrmister*'s phylogenetic position within the Haeteriinae clade, the fossil's anatomy implies a chemical strategy to become accepted or at least tolerated inside colonies (hypothetical reconstruction in *Figure 2F*), akin to modern haeteriine species that have so far been examined (*Akre, 1968*; *Lenoir et al., 2012*; *Seyfried, 1928*).

What were the Cretaceous host ants of *Promyrmister*? All ants thus far described from Burmese amber belong to stem-group Formicidae, including members of the extinct subfamily Sphecomyrminae and three other genera, *Gerontoformica*, *Myanmyrma* and *Camelomecia* that similarly lack crown-group features but are placed *incertae sedis* within Formicidae (*Barden, 2016*; *Barden, 2018*). In contrast, fossils of definitive crown-group ant subfamilies are absent, or vanishingly rare, among the thousands of ant inclusions now recovered from this amber deposit (*Barden, 2016*; *McKellar et al., 2013*) (P. Barden, personal communication). Crown-group ants are also unknown from contemporaneous Charentese amber (*Barden, 2016*). We posit that the overwhelming prevalence of stem-group ants in Burmese amber implies that they were potential hosts of *Promyrmister* (*Figure 2F*). Such a scenario entails that haeteriines may not have originated with the modern ant

groups that host them today; instead myrmecophily evolved first in stem-group ant colonies, with the beetles later switching to crown-group ants. We cannot, however, rule out an alternative scenario, that an as-yet undiscovered diversity of crown-group ants were, in fact, present in the Burmese palaeofauna, and it was these that selected for the early evolution of myrmecophily. Molecular dating indicates that crown-group ants had originated by this time (*Brady et al., 2006*; *Moreau and Bell, 2013*; *Borowiec et al., 2017a*) (see dotted lines in *Figure 3*). If present in this ancient ecosystem, perhaps their cryptic biologies limited their entrapment in amber.

Whether haeteriines evolved in stem- or crown-group ant colonies, their original hosts are presumably long-extinct. The present-day host associations of haeteriines imply that these myrmecophiles have host-switched between many modern ant lineages (*Figure 3*). The beetles have been recorded in colonies of ant species scattered across the subfamilies Dolichoderinae, Dorylinae, Formicinae, Myrmicinae and Ponerinae (*Helava et al., 1985*; *Tishechkin, 2007*) (*Figure 3*). We suggest that it is this capacity for host switching that may explain the great longevity of the clown beetle-ant symbiosis. Through host switching, the clade as a whole has circumvented potential coextinction with host ant lineages that disappeared from the Cretaceous to the present (*Barden and Grimaldi, 2016*; *Barden, 2016*). Moreover, in some cases, the beetles have radiated dramatically with certain ant groups: the vast majority of the contemporary species richness of Haeteriinae is found in taxa that have adapted to colonies of Neotropical army ants (Ecitonini), including at least 30 genera associated with *Eciton* army ants alone (*Parker, 2016*; *Helava et al., 1985*; *von Beeren and Tishechkin, 2017*; *Tishechkin, 2007*). Some of these haeteriines have remarkable adaptations for life in colonies of those nomadic ants (*Figure 4*). Neotropical army ants are thought to have begun diversifying approximately in the Oligocene (*Brady et al., 2014*; *Borowiec et al., 2017b*), implying that the bulk of haeteriine cladogenesis occurred within this window too, long after the beetles originated in the Cretaceous.

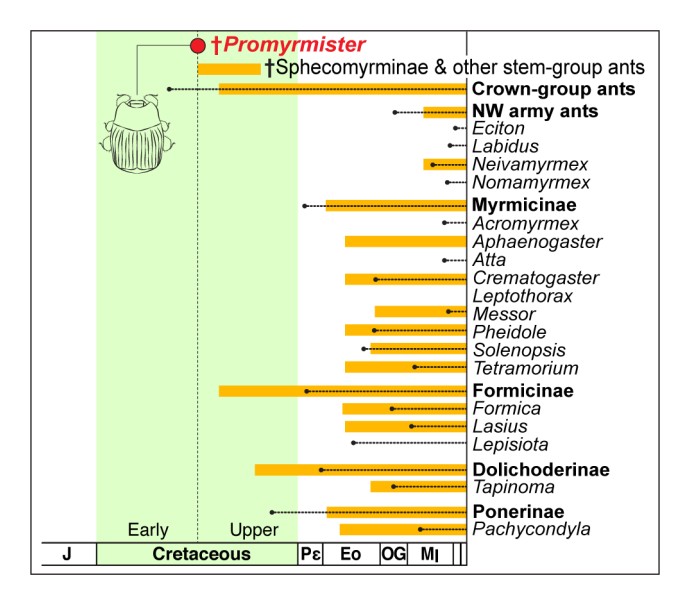

**Figure 3.** Antiquity of *Promymister* implies pervasive host switching of Haeteriinae from Cretaceous to Recent. Age of *Promyrmister* is shown (red circle). The inferred window of occurrence of stem-group ants is indicated by the top orange bar. The ages of crown-group ants as a whole, New World (NW) army ants, and other specific subfamilies and genera that are known hosts of Haeteriinae are also presented. Orange bars extend back from the Recent to the age of the earliest-known fossil; dotted lines extend back to molecularly-inferred origins of crown groups. Molecular dating implies crown-group ants existed at the same time as *Promyrmister*, but stem-group ants are the only ants so far known in Burmese and other contemporaneous ambers. All modern host ant genera are inferred to have Cenozoic origins, implying extensive host switching between the inferred Early Cretaceous origin of Haeteriinae and the present day.
DOI: https://doi.org/10.7554/eLife.44985.010

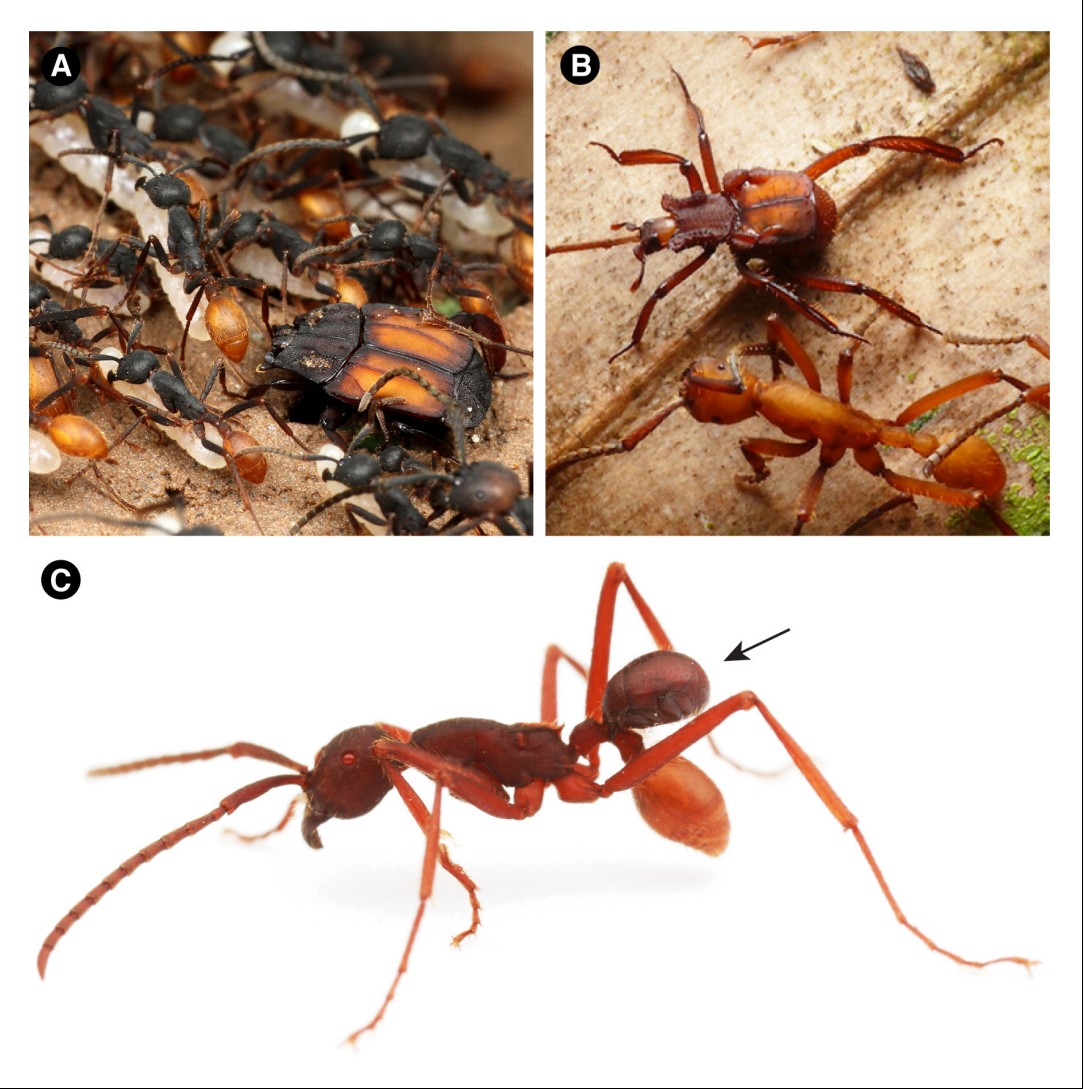

**Figure 4.** Diversity of modern Haeteriinae associated with Neotropical army ants (Ecitonini). (**A**) *Colonides* beetle walks in an emigration of *Eciton burchellii* army ants. Note the color mimicry of the host ant (Peru; photo: Taku Shimada). (**B**) *Euxenister* beetle walking alongside an *Eciton hamatum* army ant worker. The long legs facilitate grooming ants to obtain the colony odor, as well as clinging to emigrating workers (Peru; photo credit: Takashi Komatsu). (**C**) *Nymphister kronaueri* beetle phoretically attached to the petiole of an *Eciton mexicanum* army ant worker. The beetles bite onto this part of the ant's body, enabling them to migrate with the host. The beetle resembles the ant's gaster from above, potentially camouflaging the beetles to avoid predation (Costa Rica; photo credit: Daniel Kronauer).
DOI: https://doi.org/10.7554/eLife.44985.011

An ancient association between histerids and ants is further suggested by inquiline-like morphology in two other Cretaceous clown beetle fossils (*Caterino et al., 2015*; *Caterino and Maddison, 2018*), although unlike *Promyrmister*, the taxonomic affinities of these specimens are ambiguous and they are not definitive members of wholly symbiotic lineages. We infer that Haeteriinae was a relatively diverse clade by at least the beginning of the Upper Cretaceous, and likely originated and began undergoing basal cladogenesis very soon after the inferred Early Cretaceous emergence of ant eusociality (*Barden and Grimaldi, 2016*; *Grimaldi and Agosti, 2000*; *Barden, 2016*; *Brady et al., 2006*; *Moreau and Bell, 2013*; *Borowiec et al., 2017a*). A time-calibrated molecular phylogeny of Haeteriinae may provide a more precise estimate of this temporal window. However, based on such analyses for other myrmecophile taxa, rapid evolution of specialized symbiotic

phenotypes appears to be a common feature to clades of social insect symbionts (*Moore and Robertson, 2014*; *Parker and Grimaldi, 2014*), and presumably results from intense selection pressures inside colonies (*Kistner, 1979*; *Parker, 2016*). *Promyrmister* adds further support to the view that the earliest-known ants were socially complex (*Barden and Grimaldi, 2016*). Evidently, their colonies were also resource rich enough for exploitation by impostor myrmecophiles, which we conclude have been an unremitting part of ant biology. Despite their phenotypic intricacy and obligate dependency on other species, complex behavioral relationships between animals can be extraordinarily ancient, and persist over deep evolutionary time.

## Materials and methods

### Material and photography

This study is based on a single specimen of Burmese amber (CNU-008021) collected from Noije Bum, Tanaing, Kachin, Myanmar. The specimen is housed at Key Laboratory of Insect Evolution and Environmental Changes, Capital Normal University, Beijing. The holotype of the new genus and species is embedded in a cuboid amber piece. The holotype was examined under a Leica M205C dissecting microscope and photographed using a Visionary Digital BK Lab Plus system (Austin, Texas). The source images were aligned and stacked in Helicon Focus (Ukraine). Fluorescence images of the fossil were made on a Zeiss LSM 880 (with Airyscan) confocal microscope (Germany) with a 488 nm laser. Scanning electron microscopic images of *Haeterius* were obtained using a Tabletop Hitachi Microscope TM3030Plus (Japan). Morphological terminology follows *Ślipiński and Mazur (1999)*, *Zhou et al. (2018)*, and *Caterino and Tishechkin (2015)*.

### Taxon sampling, morphological characters and phylogenetic analysis

We scored *Promyrmister* for 259 external morphological characters used by *Caterino and Tishechkin (2015)* in a study investigating relationships among the tribe Exosternini, which is closely related to Haeteriinae. From the original matrix, we selected all taxa from the nearest sister clades of Haeteriinae, including 35 taxa belonging to Exosternini, including all species of *Yarmister* (apparently the closest genus to Haeteriinae; *Caterino and Tishechkin, 2015*). We also included representatives of four other tribes: Omalodini, Histerini, Hololeptini and Platysomatini, and assigned *Hister unicolor* as the primary outgroup, following *Caterino and Tishechkin (2015)*. The final taxon list is presented in *Supplementary file 2A*.

We also enlarged our data matrix by adding one more taxon (*Haeterius ferrugineus*) two more characters (260 and 261), and one more state for Character 14:

260: Epistoma: (1) without depressions receiving scapes in repose, occasionally with small depressions but without sharp arched-inwards epistomal striae; (2) with large depressions receiving scapes in repose, often defined by sharp arched-inwards epistomal striae.

261: Arched-inwards epistomal striae: (1) convergent, but separated from each other in the middle; (2) convergent, and meeting each other in the middle; (3) inapplicable.

14: Epistoma, surface: 6) deeply depressed, with lateral ridges (=raised epistomal striae) aligned with frontal stria.

The complete matrix of 46 taxa, 261 characters was constructed in Mesquite v. 3.20 (*Maddison and Maddison, 2016*); the matrix is provided in the nexus file (*Supplementary file 1*). Bayesian analysis was carried out using MrBayes 3.2.6 (*Ronquist et al., 2012*) accessed via the CIPRES Science Gateway Version 3 (*Miller et al., 2010*) (phylo.org). The Mkv model of character evolution was used with a gamma distribution, and two MCMC were executed with four chains for 100 million generations. Convergence was judged to have occurred when the standard deviation of split frequencies dropped below 0.005, and by ESS values higher than 200 in Tracer v1.7.0 (*Rambaut et al., 2018*), indicating adequate estimation of the posterior. The first 25% of trees were discarded as burn-in. We used Treeannotator (*Bouckaert et al., 2014*) to obtain the maximum clade credibility tree from post burn-in trees (ESS > 200) (*Figure 2A*), and added the estimated nodal Bayesian posterior probability (BPP) in FigTree v1.4.3 (https://github.com/rambaut/figtree/). Parsimony analysis was conducted in TNT Version 1.5 (*Goloboff and Catalano, 2016*) using Traditional Search without, and with implied weighting setting (function K = 13 in *Figure 2—figure supplements 2* and *3*). A consensus tree (*Figure 2B*; L = 1604, CI = 25, RI = 42) was obtained from four

shortest-length trees (L = 1483, CI = 28, RI = 48) and the branch support was also calculated using 10,000 bootstrap replicates. Mapping character state changes onto the tree was performed in Win-Clada (*Nixon, 2002*).

## Host ant ages

A list of haeteriine host ant genera was obtained from the literature (*Helava et al., 1985*; *Yéla-mos, 1997*; *Tishechkin, 2007*; *Lapeva-Gjonova, 2013*; *Mazur, 1981*). To estimate ages of stem-group and Recent host ant taxa in *Figure 3a*, data for earliest-known fossils were obtained from *Barden (2016)* (*Barden, 2016*; *Barden, 2018*), and molecular age estimates of crown-groups were taken from recent taxon-specific phylogenetic studies (*Borowiec et al., 2017a*; *Borowiec et al., 2017b*; *Blaimer et al., 2015*; *Ward et al., 2015*; *Ferguson-Gow et al., 2014*; *Ward et al., 2010*; *Schmidt, 2013*). Data are presented in *Supplementary file 2B*.

## Nomenclatural acts

This published work and the nomenclatural acts it contains have been registered in ZooBank, the online registration system for the International Code of Zoological Nomenclature. The ZooBank LSIDs (Life Science Identifiers) can be resolved and the associated information viewed through any standard web browser by appending the LSID to the prefix 'http://zoobank.org/'. The LSIDs for this publication are to be found at:

urn:lsid:zoobank.org:pub:4AE2E535-B2B7-4A9A-829F-FA17CB98AD9C.

The specific LSIDs for new nomenclatural acts:

Genus: urn:lsid:zoobank.org:act:8125C3AB-A6C3-41AF-A9D7-0B67BBA2ACAD
Species: urn:lsid:zoobank.org:act:56DE873C-0163–4 F94-8D09-196F20B84C57

## Data availability

All data generated or analyzed during this study are included in this published article (and its Supplementary Information files). The holotype specimen of *Promyrmister kistneri* is housed at Key Laboratory of Insect Evolution and Environmental Changes, Capital Normal University, Beijing (accession number CNU-008021).

# Acknowledgements

We are grateful to Alexey K Tishechkin (USDA), Michael S Caterino (Clemson University) and Alfred Newton (Field Museum) for their helpful advice on the placement of the fossil and to Margaret K Thayer (Field Museum) for a very thorough review of the paper. Phil Barden (New Jersey Insitute of Technology) provided invaluable insight into the ant fossil record and possible hosts of *Promyrmis-ter*. This research was supported by a Shurl and Kay Curci Foundation Research Grant, a Rita Allen Scholars Award and a Klingenstein-Simons Fellowship Award in the Neurosciences to JP, and the National Natural Science Foundation of China Grant no. 31402008 and International Postdoctoral Exchange Fellowship no. 20150064 to YLZ. Rolf Oberprieler (ANIC) and Hong Pang (Sun Yat-Sen University) helped with the fossil preparation, Lauren Ashman (ANIC) provided advice on improving the manuscript and Cate Lemann (CSIRO) provided technical assistance.

# Additional information

### Funding

| Funder | Grant reference number | Author |
| --- | --- | --- |
| National Natural Science Foundation of China | 31402008 | Yu-Lingzi Zhou |
| International Postdoctoral Exchange Fellowship | 20150064 | Yu-Lingzi Zhou |
| Rita Allen Foundation | | Joseph Parker |

| Esther A. and Joseph Klingen-stein Fund | Joseph Parker |
| --- | --- |
| Shurl & Kay Curci Foundation | Joseph Parker |
| Simons Foundation | Joseph Parker |

The funders had no role in study design, data collection and interpretation, or the decision to submit the work for publication.

### Author contributions

Yu-Lingzi Zhou, Conceptualization, Data curation, Formal analysis, Investigation, Visualization, Methodology, Writing—original draft, Writing—review and editing; Adam Ślipiński, Conceptualization, Resources, Data curation, Formal analysis, Funding acquisition, Investigation, Visualization, Methodology, Writing—original draft, Project administration, Writing—review and editing; Dong Ren, Resources, Funding acquisition, Project administration; Joseph Parker, Conceptualization, Resources, Data curation, Formal analysis, Validation, Investigation, Visualization, Methodology, Writing—original draft, Project administration, Writing—review and editing

### Author ORCIDs

Joseph Parker  http://orcid.org/0000-0001-9598-2454

### Decision letter and Author response

Decision letter https://doi.org/10.7554/eLife.44985.016
Author response https://doi.org/10.7554/eLife.44985.017

# Additional files

### Supplementary files

- Supplementary file 1. Complete matrix of 46 taxa, 261 characters constructed in Mesquite v. 3.20.
DOI: https://doi.org/10.7554/eLife.44985.012

- Supplementary file 2. Histerid taxa sampled for phylogenetic analysis, and ages of haeteriine host ant genera inferred from fossil and molecular data.
DOI: https://doi.org/10.7554/eLife.44985.013

- Transparent reporting form
DOI: https://doi.org/10.7554/eLife.44985.014

### Data availability

All data generated or analyzed during this study are included in the manuscript and supporting files. Source data for Figure 2, Figure 2 figure supplement 2 and Figure 2 figure supplement 3 are provided in Supplementary File 1.

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
