## [Decision Letter]

[Editors’ note: a previous version of this study was rejected after peer review, but the authors submitted for reconsideration. The first decision letter after peer review is shown below.]

Thank you for submitting your work entitled "A Mesozoic Clown Beetle Myrmecophile" for consideration by *eLife*. Your article has been reviewed by three peer reviewers, one of whom is a member of our Board of Reviewing Editors, and the evaluation has been overseen by a Senior Editor. The following individual involved in review of your submission has agreed to reveal their identity: Margaret Thayer (Reviewer #3).

Our decision has been reached after consultation between the reviewers. Based on these discussions and the individual reviews below, we regret to inform you that your work will not be considered further for publication in *eLife*.

The paper is a significant contribution to our understanding of the evolution of social relationships of these insect groups in its findings, and is generally well-written and well-illustrated. However some of the findings are inferred and stated very strongly, as highlighted by reviewer 2, and there are also some major issues raised by reviewer 3, concerning the diagnostic characters of the new genus, and the characters used in phylogenetic analysis.

Reviewer #1:

This is a well-written and beautifully illustrated paper describing a new taxon of fossil clown beetle Mymrecophile from the Burmese amber lagerstatte of Late Cretaceous age. The significance of the discovery lies in the fact that clown beetles are nest intruders into ant colonies, so this fossil gives a lower end date for the beginning of this type of complex inter-social adaptation in these two insect groups. From the fossil evidence the authors deduce that such obligate relationships must have evolved rapidly, and began earlier in the Cretaceous.

The perfect 3D preservation of the fossil beetle allows it to be describe in detail, and shows clear taxonomic features so taxonomy is not disputed here. It is so rare in the fossil record to get this kind of preservation that the vast amount of information coming from the Burmese amber insects makes it difficult to determine precisely when such complex social behaviours first originated (we would need other amber lagerstatten in the Jurassic and /or the Triassic to pin-point such events).

I expect the arthropod taxonomists to offer more detailed comments about the taxonomy etc. My only comments for the authors to consider are minor ones:

Results and Discussion: 'mid-Cretaceous' as anon formal date is here confusing as the age of the deposit make sit Cenomanian (technically "Upper Cretaceous"), and absolute age place sits in the top 1/3 of the Cretaceous age span, so Upper cretaceous is more preferable than using' mid-Cretaceous'.

Please consider my comments above about the 'lagerstate effect' -such well-preserved fossils can suggest biases in the fossil record, so estimating dates of origin of social behaviour may rest upon a combination of molecular dates of when each group might first evolved calibrated with the dated fossil record.

Reviewer #2:

This study reports a myrmecophilous beetle of the histerid subfamily Haeteriinae from ca. 100 million year old amber from Myanmar. This fossil is of particular interest, because it coincides with early ant evolution, suggesting that social parasitism in histerids evolved early on and has persisted until today. The manuscript is extremely well written and beautifully illustrated. However, I couldn't escape the impression that the discovery is being oversold.

1) Throughout the manuscript, the authors emphasize the point that the described beetle must have been a symbiont of stem-group ants, i.e. ants that are represented in the fossil record but only distantly related to extant ants. Their main argument is that extant crown-group ants are not represented in Burmese amber. They also argue that molecular dating analyses suggest that extant host taxa of Haeteriinae are of more recent age. But I think there are a few issues with this argument. First, the fossil record is incomplete, and absence of extant ants in the fossil record therefore doesn't mean that they weren't there. Second, there is in fact at least one record of crown-group ants in Burmese amber (the extant subfamily Aneuretinae), showing that stem-group and crown-group ants coexisted during that time (see e.g. Barden, 2017, cited by the authors). Furthermore, the most recent estimates based on molecular data date the origin of crown-group ants to ca. 103-124 MYA, i.e. several million years prior to the Burmese amber deposits (Borowiec et al., 2017; not cited by the authors). This all suggests that crown-group ants in fact co-occurred with the described beetle. The evolutionary scenario described (subsection “*Sphecomyrmister* and deep time persistence of a social symbiosis", fourth paragraph) posits that these beetles originally evolved with stem-group ants and then switched to crown-group ants later. But there is really no evidence for this. In fact, from a purely parsimony principle point of view, it seems more parsimonious that the beetles evolved to parasitize early crown-group ants, and that they can still be found associated with a subset of lineages today. If you allow for host switching anyway, then the fact that extant hosts of Haeteriinae are of younger age than the fossil beetle (Figure 3) also doesn't strengthen the argument that they would have originally come from stem-group ants. Unfortunately, the stem-group ant angle is the main selling point of the manuscript – it's reflected in Figure 2F (which I find misleading) and even the proposed genus name of the beetle (*Sphecomyrmister*), and it takes up large proportions of the Introduction and Discussion. But I would say that the support for this is rather weak.

2) A couple of other recent papers have hinted at associations between histerids and ants of similar age (Caterino, Wolf-Schwenninger and Bechly, 2015, and Maddison and Caterino, 2008; discussed in the last paragraph of the subsection “*Sphecomyrmister* and deep time persistence of a social symbiosis”), and I think that takes away from the novelty of the current discovery. Beetle fossils could in principle be associated with ants in three ways I guess. The most convincing would be inclusions that contain both the beetles and the host ants which, as far as I know, have not been found. The second most compelling evidence comes from morphological adaptations that are restricted to inquilines – this is the case both for this fossil and the previous fossils. The third and, by itself, arguably most circumstantial is phylogenetic placement of the fossil. The fossil described here can be phylogenetically placed within a group of obligately myrmecophilous beetles, which of course is very nice. But I don't think that the evidence for myrmecophily is so much stronger here compared to the previous fossils. So even though I must admit that the packaging is much more compelling here, I therefore fail to see the major advancement in our knowledge of myrmecophile evolution.

Reviewer #3:

The new fossil taxon described in this paper represents an exciting addition to our knowledge of both the beetle family Histeridae and, more broadly, evolution of myrmecophily in beetles. The authors describe and illustrate the new fossil well and provide convincing justification for placing the new taxon within the haeteriine Histeridae. Since all known extant Haeteriinae are obligate myrmecophiles, the morphology-based placement supports the inference that *Sphecomyrmister* was also a myrmecophile, although in a few places the authors tend toward presenting this as a fact rather than an inference.

Nevertheless, I have some concerns about the phylogenetic analysis. Although the analysis itself was carried out in reasonable fashion, simple adoption of a data matrix from a paper (Caterino and Tishechkin, 2015) that focused on a different part of the Histeridae (Histerinae: Exosternini) is problematic. The original matrix included a few representatives of Haeteriinae, along with hundreds of other taxa belonging to the focal group of that paper and an assortment of other Histeridae as outgroups. The taxon sub-sampling in the current manuscript from the larger Caterino and Tishechkin, 2015 matrix seems reasonable at first glance, but it is not clear how or why they chose particular Exosternini genera (aside from Yarmister), and on further consideration I am concerned by two fundamental aspects of their analysis.

First, although understanding the relationships of *Sphecomyrmister* within Haeteriinae is presumably a major focus of the paper, the authors did not add to the matrix any characters that might be suitable or necessary for resolving those relationships. Especially considering the specialized morphology of Haeteriinae, surely there are relevant characters that were not included in Caterino and Tishechkin, 2015's analysis of other Histeridae, such as those cited in the current manuscript to (in combination) separate *Sphecomyrmister* from all other Haeteriinae. (I admit I did not examine Caterino and Tishechkin, 2015's long character list in detail to search for those.)

Second, the authors did not include representatives of any of the three largest and most heavily sampled genera in Caterino and Tishechkin, 2015's paper (Phelister, Operclipygus, and Baconia), and for some reason all but one of the Exosternini genera included are Old World rather than New World taxa (a point not mentioned), even though the focus of Caterino and Tishechkin, 2015's paper was Neotropical Exosternini and nearly all Haeteriinae are Neotropical. Although selecting single species from the large genera might have been challenging (there appears to be substantial variation within each genus), I found that completely leaving them out led to partly spurious results. Specifically, Figure 2—figure supplement 2 seems to show as unique apomorphies for *Sphecomyrmister* (I think-the character state numbers are blurry) the states 12-2, 112-2, 113-2, and 114-2; this is puzzling since the (new) genus obviously was not included in Caterino and Tishechkin, 2015's data but the characters were. For the latter three characters, Caterino and Tishechkin, 2015, said the modified gland openings involved occur only in some species groups of Operclipygus (Exosternini). Clearly, then, they are not globally unique to *Sphecomyrmister*, but appear as such in the present analysis because no Operclipygus were included! Similarly, Caterino and Tishechkin, 2015, illustrated state 12-2 with figures of two species of Baconia (Exosternini), so again this is not truly a unique apomorphy of *Sphecomyrmister*. Figure 2—figure supplement 2 does not show any unique apomorphies for Haeterius, so reciprocal monophyly of that and *Sphecomyrmister* do not seem to be supported. Mirroring that problem in the text, although the authors list a combination of characters to separate *Sphecomyrmister* from all other Haeteriinae, and others to show its close relationship to Haeterius, I could not find a clear statement of what separates those two genera. This is unsatisfactory from two standpoints: 1) compliance with the [ICZN] Code requirement for a statement purporting to distinguish the new taxon and 2) supporting the authors' evolutionary contention that *Sphecomyrmister* represents an extinct Cretaceous lineage.

---

## [Author Response]

[Editors’ note: the author responses to the first round of peer review follow.]

Thank you so much for your excellent feedback on our paper. We have taken every effort to improve the new version based on your constructive comments. We have briefly summarized your reviews, and our responses to them:

Reviewer 1 recognizes the novelty and valuable contribution of our paper. Reviewer 1 notes that due to the incompleteness of the fossil record, it is “difficult to determine precisely when such complex social behaviours first originated”. Our fossil provides a minimum age, but future molecular dating of groups like Haeteriinae will be illuminating. We agree, and have included this suggestion in the revised manuscript, along with citations to papers that have done exactly this for other myrmecophile clades.

Reviewer 2 agrees that our fossil discovery is “of particular interest, because it coincides with early ant evolution” but questions whether stem-group ants were likely hosts. In the revised paper, we present our original scenario together with its tantalizing alternative: that an as-yet undiscovered diversity of crown-group ants could have lived in the Burmese palaeofauna, and been potential hosts. Unlike reviewers 1 and 3, reviewer 2 also argues that previously-described ‘myrmecophile-looking’ histerids reduce our paper’s novelty. We respectfully disagree: prior speculation cannot match the discovery of hard evidence. Our paper uncovers the earliest definitive evidence of Cretaceous myrmecophily, contemporaneous with the earliest-known eusocial ants.

Reviewer 3 recognizes the novelty and valuable contribution of our paper. Reviewer 3 has thoroughly studied our phylogenetic analysis and raised several legitimate criticisms. In the revised paper, we have addressed these criticisms for a more solid phylogenetic analysis and taxonomic placement of the fossil. We have also attended to numerous minor errors that reviewer 3 highlighted.

The paper is a significant contribution to our understanding of the evolution of social relationships of these insect groups in its findings, and is generally well-written and well-illustrated. However some of the findings are inferred and stated very strongly, as highlighted by reviewer 2, and there are also some major issues raised by reviewer 3, concerning the diagnostic characters of the new genus, and the characters used in phylogenetic analysis.Reviewer #1:[…] The perfect 3D preservation of the fossil beetle allows it to be describe in detail, and shows clear taxonomic features so taxonomy is not disputed here. It is so rare in the fossil record to get this kind of preservation that the vast amount of information coming from the Burmese amber insects makes it difficult to determine precisely when such complex social behaviours first originated (we would need other amber lagerstatten in the Jurassic and /or the Triassic to pin-point such events).Such well-preserved fossils can suggest biases in the fossil record, so estimating dates of origin of social behaviour may rest upon a combination of molecular dates of when each group might first evolved calibrated with the dated fossil record.

Thank you for your wonderful comments. We agree with your major point that inferring more precisely when myrmecophily in haeteriines evolved is still murky. Our fossil implies the Early Cretaceous at the latest, but molecular dating of a comprehensive phylogeny of these beetles would provide more resolution, or at least a more precise inference. Such efforts have been used for certain clades of myrmecophile staphylinids, as well as paussine carabids. In the new paper, we make mention of this, and argue that creating a dated phylogeny of haeteriinae, (and more broadly, Histeridae) is particularly timely given this discovery of our fossil and its implications for symbiotic relationships of early ants.

Reviewer #2:This study reports a myrmecophilous beetle of the histerid subfamily Haeteriinae from ca. 100 million year old amber from Myanmar. This fossil is of particular interest, because it coincides with early ant evolution, suggesting that social parasitism in histerids evolved early on and has persisted until today. The manuscript is extremely well written and beautifully illustrated. However, I couldn't escape the impression that the discovery is being oversold.1) Throughout the manuscript, the authors emphasize the point that the described beetle must have been a symbiont of stem-group ants, i.e. ants that are represented in the fossil record but only distantly related to extant ants. Their main argument is that extant crown-group ants are not represented in Burmese amber. They also argue that molecular dating analyses suggest that extant host taxa of Haeteriinae are of more recent age. But I think there are a few issues with this argument. First, the fossil record is incomplete, and absence of extant ants in the fossil record therefore doesn't mean that they weren't there. Second, there is in fact at least one record of crown-group ants in Burmese amber (the extant subfamily Aneuretinae), showing that stem-group and crown-group ants coexisted during that time (see e.g. Barden, 2017, cited by the authors). Furthermore, the most recent estimates based on molecular data date the origin of crown-group ants to ca. 103-124 MYA, i.e. several million years prior to the Burmese amber deposits (Borowiec et al., 2017; not cited by the authors). This all suggests that crown-group ants in fact co-occurred with the described beetle. The evolutionary scenario described (subsection “Sphecomyrmister and deep time persistence of a social symbiosis", fourth paragraph) posits that these beetles originally evolved with stem-group ants and then switched to crown-group ants later. But there is really no evidence for this. In fact, from a purely parsimony principle point of view, it seems more parsimonious that the beetles evolved to parasitize early crown-group ants, and that they can still be found associated with a subset of lineages today. If you allow for host switching anyway, then the fact that extant hosts of Haeteriinae are of younger age than the fossil beetle (Figure 3) also doesn't strengthen the argument that they would have originally come from stem-group ants.

Thank you for raising this point. In our original manuscript we felt compelled to point out that stem-group subfamilies seemed the most likely hosts, because these are the only ones known for definite in Burmese amber (see separate comment below about the aneuretine). We are aware that molecular dating studies imply that crown-group ants may have existed at this time, but because they are simply not known in thousands of ant inclusions in Burmese amber, we felt it highly unlikely that their myrmecophiles would make an appearance but they would not.

However, we acknowledge that the likelihood of whether stem- or crown-group ants were the beetle’s hosts could be judged equivocal, depending on one’s preference for data type. In our view, the overwhelming predominance of stem-group ant fossils in Burmese amber would imply that these were the hosts – it is not a stretch to think that stem group ant colonies were resource rich enough to select for the initial evolution of myrmecophily. Conversely, based on reviewer 2’s entirely reasonable suggestion, molecular dating implies that crown-group ants had evolved by 99MYA, so could also legitimately be hosts. In the revised paper, we present both hypotheses as valid, although convey our preference for the former scenario, which we think is not unreasonable based on the lack of crown-group ant subfamilies in Burmite. Accordingly, we have modified the taxon’s name to accommodate both hypotheses (we have adopted *Promyrmister* – “early ant hister”). We have kept Figure 3, but modified it with molecular dates from Borowiec et al., 2017, and now use this figure to explain the two alternative hypotheses. Modifying the paper in this way does not detract from the impact or novelty of the story – quite the opposite – it raises two, equally fascinating possibilities: either stem-group ants were the beetle’s hosts, or the beetle alludes to an as-yet undiscovered diversity of crown-group ants in the Burmese palaeofauna.

Second, there is in fact at least one record of crown-group ants in Burmese amber (the extant subfamily Aneuretinae), showing that stem-group and crown-group ants coexisted during that time (see e.g. Barden, 2017, cited by the authors).

Please note that the identity of this specimen, *Burmomyrma rossi*, is highly contentious. Even Dlussky, the taxon’s author, expressed doubt that this specimen was an aneuretine. According to Barden, 2017:

“Because the sole specimen (an alate female) is missing the entire head and portions of the mesosoma, Dlussky, 1996, was initially equivocal in his assignment of Burmomyrma to Aneuretinae, stating the “systematic position could not be determined reliably due to poor preservation of the only specimen known”. Tentative aneuretine placement was based on highly reduced forewing venation, curved sting, a single segmented petiole, and a gaster without constrictions. This particular assortment of characters cannot be used to assign *Burmomyrma rossi* to any subfamily with confidence, particularly as the wing venation described is not shared by other known aneuretines (Boudinot, 2015).”

Moreover, a recent work concluded that *Burmomyrma rossi* is probably not an ant at all, but a chrysidid wasp (See: Lucena, and Melo, Cretaceous Research 89, 279–291). Consequently, there are no definitive crown-group ants in Burmese amber.

2) A couple of other recent papers have hinted at associations between histerids and ants of similar age (Caterino, Wolf-Schwenninger and Bechly, 2015 and Maddison and Caterino, 2008; discussed in the last paragraph of the subsection “Sphecomyrmister and deep time persistence of a social symbiosis”), and I think that takes away from the novelty of the current discovery. Beetle fossils could in principle be associated with ants in three ways I guess. The most convincing would be inclusions that contain both the beetles and the host ants which, as far as I know, have not been found. The second most compelling evidence comes from morphological adaptations that are restricted to inquilines – this is the case both for this fossil and the previous fossils. The third and, by itself, arguably most circumstantial is phylogenetic placement of the fossil. The fossil described here can be phylogenetically placed within a group of obligately myrmecophilous beetles, which of course is very nice. But I don't think that the evidence for myrmecophily is so much stronger here compared to the previous fossils. So even though I must admit that the packaging is much more compelling here, I therefore fail to see the major advancement in our knowledge of myrmecophile evolution.

We respectfully disagree. Previously-described fossil histerids with myrmecophile-like morphology cannot compete with the impact of our story. Such a suggestion equates the discovery of scientific evidence with the mere speculation that preceded the discovery. Morphology alone does not meet the burden of proof for myrmecophily – there are numerous extant insect groups that have a somewhat myrmecophile-like appearance but are not myrmecophiles (e.g. in Carabidae: the subfamily Rhysodinae and the scaritine genera *Solenogenys* and *Salcedia*; in Staphylinidae: Falagriini such as *Myrmecocephalus*, many Pselaphinae such as *Brachygluta abdominalis*). In contrast, a species with myrmecophilous adaptations that additionally belongs to a clade composed entirely of obligate myrmecophiles leaves little doubt.

Prior to our paper, definitive myrmecophiles were unknown from the entire Cretaceous, let alone from the same deposit as the earliest-known ants. The literature is sprinkled with speculative examples of Cretaceous insects claimed to have morphology suggestive of myrmecophily or termitophily (e.g. a supposed myrmecophiline cricket in the Crato Formation (Martins-Neto, 1991; Parker and Grimaldi, 2014); a Lebanese amber scarabaeoid (Crowson, 1981)). The two histerids mentioned are but two more such examples. Yet, the whole notion of Cretaceous myrmecophily, and early ant colony infiltration, has remained entirely speculative due to lack of any hard evidence in the form of a specimen belonging to wholly myrmecophilous group.

Our paper now uncovers a bona fide, anatomically specialized myrmecophile from the crown-group of a modern clade composed entirely of such creatures. The fossil satisfies the essential burden of proof of myrmecophily. That such an organism could have existed in the earliest-known ant societies in the Cretaceous is remarkable. The fossil is a member of one of the largest-known modern radiations of myrmecophiles, illuminating the deep ancestry of the social symbiosis between clown beetles and ants. The implications are broad for the evolution of social interactions, early colony formation in ants, and symbiotic relationships between animals more generally. Our paper is thus of special value compared to the speculative works that preceded it.

Crowson, R. A. (1981). The Biology of the Coleoptera. London: Academic Press.

Martins-Neto, R. G. (1991). Sistemática dos Ensifera (Insecta, Orthopteroida) da formação Santana, Cretáceo Inferior do Nordeste do Brasil. Acta Geologica Leopoldensia 32, 5–160.

Reviewer #3:The new fossil taxon described in this paper represents an exciting addition to our knowledge of both the beetle family Histeridae and, more broadly, evolution of myrmecophily in beetles. The authors describe and illustrate the new fossil well and provide convincing justification for placing the new taxon within the haeteriine Histeridae. Since all known extant Haeteriinae are obligate myrmecophiles, the morphology-based placement supports the inference that Sphecomyrmister was also a myrmecophile, although in a few places the authors tend toward presenting this as a fact rather than an inference.

In the new paper we have changed the language in places to avoid appearing to do this.

Nevertheless, I have some concerns about the phylogenetic analysis. Although the analysis itself was carried out in reasonable fashion, simple adoption of a data matrix from a paper (Caterino and Tishechkin 2015) that focused on a different part of the Histeridae (Histerinae: Exosternini) is problematic. The original matrix included a few representatives of Haeteriinae, along with hundreds of other taxa belonging to the focal group of that paper and an assortment of other Histeridae as outgroups. The taxon sub-sampling in the current manuscript from the larger Caterino and Tishechkin, 2015 matrix seems reasonable at first glance, but it is not clear how or why they chose particular Exosternini genera (aside from Yarmister), and on further consideration I am concerned by two fundamental aspects of their analysis.

We believe the adoption of the Caterino and Tishechkin, 2015 data matrix was reasonable and justifiable. The most recent molecular phylogenies of Histeridae (Caterino and Vogler, 2002, McKenna et al., 2015) have recovered Haeteriinae as a sister group to Histerinae. Both Caterino and Tishechkin, 2015, and McKenna et al., 2015, recovered a monophyletic Haeteriinae, which is represented in our taxon sampling by genera belonging to all three recognized tribes of this subfamily. Our a priori assessment of the fossil was that it was a haeteriine, so we evaluated this placement phylogenetically, adopting the Caterino and Tishechkin, 2015 morphological data set due to its empirically demonstrated ability to resolve Haeteriinae monophyly. The original taxon sampling in Caterino and Tishechkin, 2015 is vast and exceeded what we needed for our intended purpose. So, the haeteriines and a subset of outgroups were used. The taxonomic scope of our analysis was, we felt, sufficient for just seeing whether the fossil was recovered in Haeteriinae, which our a priori assessment had implied. However, we agree that a more objective judgment of the placement of the fossil taxon could be achieved, so we have performed a revised analysis. We scored one more Haeterius species ourselves, and included at least one species (type species of the genus if available) of the genera of Exosternini belonging to the sister clades of Haeteriinae, and added one species of Baconia, three species of Operclipygus, all species of Yarmister, and New and Old World representatives of Hypobletus from Caterino and Tishechkin, 2015. We believe this represents a sufficient sampling of histerid outgroups for the sole purpose of testing the fossil’s placement within the Haeteriinae clade.

First, although understanding the relationships of Sphecomyrmister within Haeteriinae is presumably a major focus of the paper, the authors did not add to the matrix any characters that might be suitable or necessary for resolving those relationships. Especially considering the specialized morphology of Haeteriinae, surely there are relevant characters that were not included in Caterino and Tishechkin, 2015's analysis of other Histeridae, such as those cited in the current manuscript to (in combination) separate Sphecomyrmister from all other Haeteriinae. (I admit I did not examine Caterino and Tishechkin, 2015's long character list in detail to search for those.)

The morphological treatment in the Caterino and Tishechkin, 2015 paper is amazingly detailed and comprehensive. It is hard to find structures or features of adult beetles that are not covered there. Most of the characters from the Caterino and Tishechkin, 2015 paper can be observed and coded from our amber fossil, and the character set was evidently comprehensive enough to resolve a monophyletic Haeteriinae, both in our analysis in the original Caterino and Tishechkin, 2015 paper. Furthermore, to address your comment more precisely, while some haeteriines are extremely anatomically specialized for myrmecophily, many are not so dramatically modified. For example, the external morphology of many Exosternini and Haeteriinae is very similar, which has led to misplacement of several genera in the past (e.g., Kaszabister, Yarmister, Tarsilister). This means that potential new, haeteriine-relevant characters are quite hard to find. Despite this, we have found and added two more characters (no. 260, 261) and one more state (6) to character 14 to distinguish Haeteriinae from outgroups. We hope this addresses your comment adequately.

Second, the authors did not include representatives of any of the three largest and most heavily sampled genera in Caterino and Tishechkin, 2015's paper (Phelister, Operclipygus, and Baconia), and for some reason all but one of the Exosternini genera included are Old World rather than New World taxa (a point not mentioned), even though the focus of Caterino and Tishechkin, 2015's paper was Neotropical Exosternini and nearly all Haeteriinae are Neotropical. Although selecting single species from the large genera might have been challenging (there appears to be substantial variation within each genus), I found that completely leaving them out led to partly spurious results.

As mentioned above, in the revised analysis we have included at least one species (type species of the genus if available) of the genera of Exosternini belonging to the sister clades of Haeteriinae, and added several species of the variable genera Operclipygus and Yarmister, as well as New and Old World representatives of Hypobletus, and one species of Baconia examined by Caterino and Tishechkin, 2015. For a more robust representation of the putuative sister genus of *Promyrmister*, we have added a Palaearctic species of Haeterius (*H. ferrugineus*) the type species of the genus, and scored it for all the characters.

Specifically, Figure 2—figure supplement 2 seems to show as unique apomorphies for Sphecomyrmister (I think-the character state numbers are blurry) the states 12-2, 112-2, 113-2, and 114-2; this is puzzling since the (new) genus obviously was not included in Caterino and Tishechkin, 2015's data but the characters were. For the latter three characters, Caterino and Tishechkin, 2015, said the modified gland openings involved occur only in some species groups of Operclipygus (Exosternini). Clearly, then, they are not globally unique to Sphecomyrmister, but appear as such in the present analysis because no Operclipygus were included! Similarly, Caterino and Tishechkin, 2015, illustrated state 12-2 with figures of two species of Baconia (Exosternini), so again this is not truly a unique apomorphy of Sphecomyrmister. Figure 2—figure supplement 2 does not show any unique apomorphies for Haeterius, so reciprocal monophyly of that and Sphecomyrmister do not seem to be supported.

We apologize, but the specific apomorphies shown in Figure 2—figure supplement 2, were not easily visible, e.g. (112-2) and (113-2) and were misread by the reviewer because of the low quality of the file. We have supplied a higher resolution image of this tree in the revised paper. To be clear, currently fifteen unambiguous apomophies have been found for *Promyrmister*: 2-2, 5-2, 11-3, 12-2, 13-2, 14-6, 56-4, 109-2, 111-3, 112-3, 112-3, 124-1, 125-1, 135-2, and 261-2. Most of them appear as homoplasious states but two of them: (14-6) (epistoma, surface deeply depressed, with epistomal striae joined basally with frontal stria), and (261-2) epistomal striae anteriorly arching-inwards, meeting each other at the middle, only appear in *Promyrmister*.

Mirroring that problem in the text, although the authors list a combination of characters to separate Sphecomyrmister from all other Haeteriinae, and others to show its close relationship to Haeterius, I could not find a clear statement of what separates those two genera. This is unsatisfactory from two standpoints: 1) compliance with the [ICZN] Code requirement for a statement purporting to distinguish the new taxon and 2) supporting the authors' evolutionary contention that Sphecomyrmister represents an extinct Cretaceous lineage.

We have now clarified in the diagnosis section that *Promyrmister* specifically differs from the putatively closely related Haeterius in having deep epistomal depressions (compare Figure 1—figure supplement 1B, C to Figure 2—figure supplement 1B), epistomal striae carinate and convergent medially (Figure 1—figure supplement 1B, C) and paddle-shaped protibia with large apical spur (Figure 1E, F). See Results and Discussion for a complete description of the new genus and species, as well as a discussion of the taxon’s systematic placement in Histeridae.